# Pathogenic *Vibrio* Species Are Associated with Distinct Environmental Niches and Planktonic Taxa in Southern California (USA) Aquatic Microbiomes

Rachel E. Diner,[a,b,c] Drishti Kaul,[c] Ariel Rabines,[a,c] Hong Zheng,[c] Joshua A. Steele,[d] John F. Griffith,[d] Andrew E. Allen[a,c]

aScripps Institution of Oceanography, University of California San Diego, La Jolla, California, USA
bDepartment of Pediatrics, University of California San Diego, La Jolla, California, USA
cMicrobial and Environmental Genomics Group, J. Craig Venter Institute, La Jolla, California, USA
dSouthern California Coastal Water Research Project, Costa Mesa, California, USA

**ABSTRACT** Interactions between vibrio bacteria and the planktonic community impact marine ecology and human health. Many coastal *Vibrio* spp. can infect humans, representing a growing threat linked to increasing seawater temperatures. Interactions with eukaryotic organisms may provide attachment substrate and critical nutrients that facilitate the persistence, diversification, and spread of pathogenic *Vibrio* spp. However, vibrio interactions with planktonic organisms in an environmental context are poorly understood. We quantified the pathogenic *Vibrio* species *V. cholerae*, *V. parahaemolyticus*, and *V. vulnificus* monthly for 1 year at five sites and observed high abundances, particularly during summer months, with species-specific temperature and salinity distributions. Using metabarcoding, we established a detailed profile of both prokaryotic and eukaryotic coastal microbial communities. We found that pathogenic *Vibrio* species were frequently associated with distinct eukaryotic amplicon sequence variants (ASVs), including diatoms and copepods. Shared environmental conditions, such as high temperatures and low salinities, were associated with both high concentrations of pathogenic vibrios and potential environmental reservoirs, which may influence vibrio infection risks linked to climate change and should be incorporated into predictive ecological models and experimental laboratory systems.

**IMPORTANCE** Many species of coastal vibrio bacteria can infect humans, representing a growing health threat linked to increasing seawater temperatures. However, their interactions with surrounding microbes in the environment, especially eukaryotic organisms that may provide nutrients and attachment substrate, are poorly understood. We quantified three pathogenic *Vibrio* species monthly for a duration of 1 year, finding that all three species were abundant and exhibited species-specific temperature and salinity distributions. Using metabarcoding, we investigated associations between these pathogenic species and prokaryotic and eukaryotic microbes, revealing genus and amplicon sequence variant (ASV)-specific relationships with potential functional implications. For example, pathogenic species were frequently associated with chitin-producing eukaryotes, such as diatoms in the genus *Thalassiosira* and copepods. These associations between high concentrations of pathogenic vibrios and potential environmental reservoirs should be considered when predicting infection risk and developing ecologically relevant model systems.

**KEYWORDS** amplicon sequence variants, chitin, diatom, metabarcoding, metagenomics, rRNA, vibrio

Address correspondence to Andrew E. Allen, aallen@ucsd.edu.

Pathogenic vibrio species are associated with distinct environmental niches and planktonic taxa in Southern California aquatic microbiomes. Pathogenic species were frequently associated with chitin-producing eukaryotes, such as diatoms and copepods

Coastal bacterial *Vibrio* species can cause severe human infections, which are an emerging international health concern linked to rising global temperatures. *Vibrio cholerae*, the causative agent of the disease cholera, infects millions of people each

year, killing thousands, and is typically spread through ingesting contaminated drinking water (1). Two other species of major concern are *V. parahaemolyticus* and *V. vulnificus*, which can cause severe wound infections, septicemia, and gastroenteritis from ingesting vibrio-colonized seafood (2). Although many strains of these species are not harmful to humans, some contain genes linked to increased virulence (3), and to date, several strains with pandemic potential have been identified. Furthermore, innocuous strains may become virulent and/or antibiotic resistant via horizontal gene transfer, as many infection-related genes are mobile (3, 4). At least a dozen additional *Vibrio* species can infect humans or animals, including aquaculture species such as shellfish. Climate change may exacerbate infections, as increasing air and water temperatures facilitate increased metabolic growth capacity and temporal and geographic range expansion of *Vibrio* spp. pathogens (5–7). Furthermore, *V. cholerae* epidemics have been linked to global temperature rise on decadal scales, serving as an important case study for understanding the link between the environment and human disease (8–10).

Coastal communities are highly productive environments; diverse and abundant populations of microbes and multicellular organisms are supported by primary productivity driven by ample nutrient availability. These communities are subject to frequent and extreme changes in environmental conditions, including fluctuations in temperature, salinity, and dissolved oxygen. In this context, *Vibrio* spp. attach to and form biofilms on particles and eukaryotic organisms, living and dead (11–14), to better acquire carbon and nutrients and avoid environmental stress. Because of this, cooccurring organisms can act as a substrate for pathogenic vibrio proliferation and potentially as vectors for transporting species to different geographic areas or causing human disease. For example, pathogenic vibrios frequently attach to chitin-producing zooplankton, such as copepods, and both high copepod abundances and high concentrations of phytoplankton are linked to cholerae epidemics, particularly in warm environments (9, 10). Attaching to these eukaryotes can also stimulate bacterial competition and horizontal gene transfer in cocolonizing *Vibrio* spp. (15, 16), which may spread virulence and antibiotic resistance genes among populations. Therefore, *Vibrio* spp. interactions with the planktonic community have implications for both coastal ecology and human health.

Vibrio interactions with prokaryotic and eukaryotic organisms are likely species-specific. Pathogenic species possess unique functional traits and often occupy distinct environmental niches driven by temperature, salinity, and other biotic and abiotic factors (reviewed in reference 17). Virulence mechanisms are also species and/or strain dependent (18). Despite this, total quantities of *Vibrio* spp. are frequently used to infer ecological associations and human health risks. Likewise, eukaryotes are often grouped into broad categories. For example, phytoplankton are often quantified and characterized based on bulk chlorophyll *a* concentration, a proxy for the abundance and biomass of photosynthetic organisms, or at broad taxonomic levels (e.g., class or class-specific pigments). But physiological and ecological differences at lower taxonomic ranks may have functional consequences for interactions; for example, some diatom genera exude chitin while others may not (19, 20), and algae host distinct bacterial communities (21). These associations are typically investigated using observational techniques; however, recent advances in *Vibrio* spp. quantification and microbial community characterization (i.e., high-throughput sequencing) of both prokaryotes and eukaryotes now enable a deeper understanding of vibrio microbial ecology in the context of environmental drivers and biological interactions. Few prior studies have utilized this technology to investigate human-pathogenic *Vibrio* species (22, 23), and these place emphasis on prokaryotic associations and/or do not quantify the vibrios being investigated.

To address this research gap, we quantified the pathogenic *Vibrio* species *V. cholerae*, *V. parahaemolyticus*, and *V. vulnificus* for 1 year at five coastal sites in Southern California and used metabarcoding to characterize the cooccurring prokaryotic and eukaryotic communities. Using a subset of these same water samples, we conducted

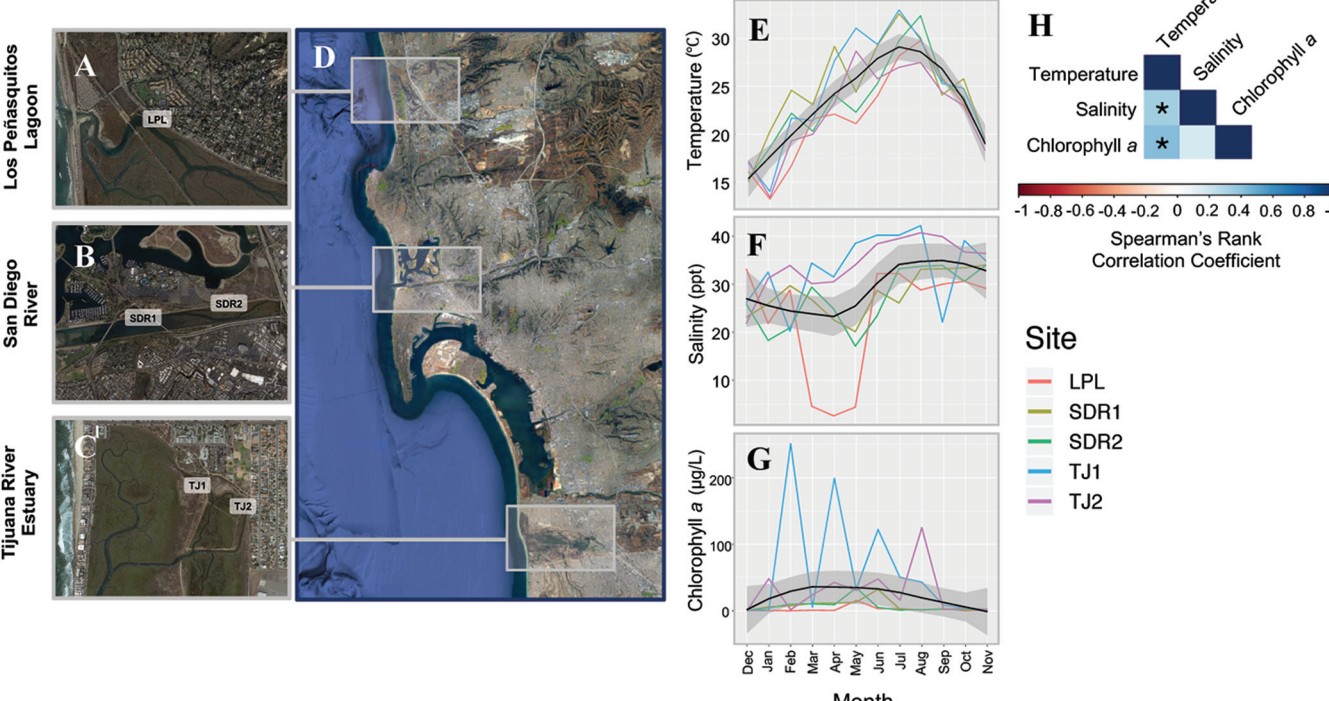

**FIG 1** Location of the sampling sites and environmental conditions at the time of sampling. Locations are mapped in the context of the San Diego region using Google Earth (maps were created using Google maps, open source). Site abbreviations are as follows: (A) LPL, Los Peñasquitos Lagoon; (B) SDR, San Diego River (site 1 and 2); and (C) TJ, Tijuana River Estuary (site 1 and 2), with (D) depicting the regional context. Environmental conditions, including temperature (E), salinity (F), and chlorophyll *a* (G), a proxy for photosynthetic organism abundance, were measured monthly at each site (LPL, red; SDR1, mustard; SDR2, green; TJ1, blue; TJ2, purple) for 1 year from December 2015 to November 2016. (H) Spearman rank correlations identified relationships between environmental variables, where values closer to 1 (dark blue) are positive correlations, and values closer to −1 (dark red) are negative correlations, and an asterisk (*) represents significant correlations ($P < 0.05$).

shotgun metagenomic sequencing of enriched vibrio bacterial cultures to identify genes of interest to human health (e.g., virulence and antibiotic resistance genes) and assessed the diversity of potentially pathogenic *Vibrio* species, allowing us to extend our knowledge of these vibrio populations beyond targeted quantification and amplicon-sequencing methods. We investigated the influence of spatial, temporal, and environmental variation on the abundance and distribution of individual pathogenic *Vibrio* species and their cooccurring coastal microbiomes. Additionally, we examined the occurrence of important species-level, or amplicon sequence variant (ASV)-level, associations and used this information to assess whether current laboratory models for studying vibrio-plankton interactions are ecologically relevant.

## RESULTS

**Sampling sites and environmental conditions.** Samples were collected monthly at five San Diego County sites (Fig. 1A to D) ($N = 60$). These samples spanned a wide range of temperatures (13.2 to 33°C) and salinities (2.6 to 42.4 ppt). Chlorophyll *a* concentrations were highly variable (Fig. 1G), which may represent variations in both system productivity and heterogeneity of available particle substrates. Salinity and chlorophyll *a* were positively associated with temperature (Fig. 1H).

**Planktonic community composition, diversity, and environmental drivers.** Profiling the eukaryotic community using 18S rRNA amplicon gene sequencing ($N = 60$ samples) revealed ∼17,000 ASVs representing phytoplankton, heterotrophic protists, and small multicellular eukaryotes, such as copepods. Diatoms were the most common eukaryotes, comprising ∼28% of 18S reads (Fig. 2A), and were particularly abundant at the Tijuana River Estuary (TJ) sites, frequently representing greater than 75% of 18S reads. Other abundant groups included unicellular *Spirotrichea* ciliates, photosynthetic

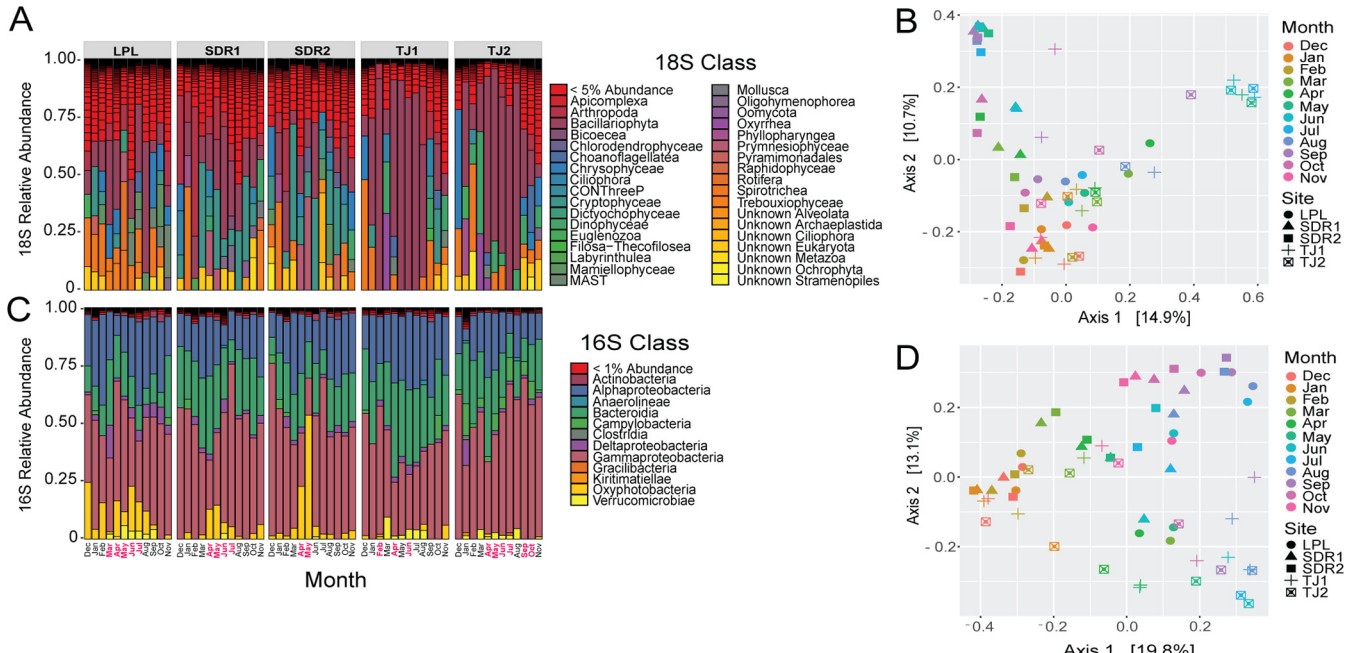

**FIG 2** Taxonomic composition and PCoA plots of eukaryotic (18S) and bacterial and archaeal (16S) communities. Taxonomic composition of 18S (A) communities by class and PCoA plot (B) based on Bray-Curtis dissimilarity. Similarly, (C) depicts the taxonomic composition of 16S communities by class, and (D) depicts a PCoA plot. For the taxonomy bar plots (A) and (C), months in red font on the *x* axis are months where >1,000 copies/ml of any pathogenic *Vibrio* species was detected by ddPCR, and cutoffs of <1% and <5% abundance represent taxa cumulative abundance across all samples.

*Cryptophyceae*, and chitin-producing zooplankton, such as copepods. There were also many low-abundance classes; at some sites, more than half of the classes were taxa comprising less than 5% relative abundance. Both observed ASVs and Shannon diversity were significantly different between months and sites (Fig. S1 in the supplemental material), and higher temperature and chlorophyll *a* were associated with lower alpha diversity. Beta diversity was strongly influenced by site and month (Fig. 2B), which accounted for 16% and 30% of eukaryotic community variance, respectively (see the data on figshare at https://dx.doi.org/10.6084/m9.figshare.13653590). Environmental variables significantly impacted variance to a lesser extent; temperature, salinity, and chlorophyll *a* explained 7%, 4%, and 4% of the variance, respectively (Fig. S2).

Among the bacterial and archaeal community, 16S rRNA gene sequencing (*N* = 60 samples) resulted in ~30,000 ASVs. *Vibrio* species comprised 0.03 to 4.9% of the 16S community. A few major bacterial classes dominated community composition, including *Gammaproteobacteria* (encompassing *Vibrio* spp.), *Bacteroidia*, and *Alphaproteobacteria* (Fig. 2C). Los Peñasquitos Lagoon (LPL) and San Diego River (SDR) sites had sizeable populations of *Oxyphotobacteria* (phylum *Cyanobacteria*), while other prominent classes included *Campylobacteria* and *Verrucomicrobia*. Sites and months did not significantly differ in alpha diversity based on Shannon diversity or observed ASVs (Fig. S3), while higher temperature and chlorophyll *a* were associated with fewer observed ASVs. Site and month also drove 16S community similarity (Fig. 2D), explaining 17% and 39% of the community variance, respectively. Likewise, environmental variables explained much of the additional variance, including temperature (12%), salinity (8%), and chlorophyll *a* (4%) (Fig. S2).

For both 16S and 18S communities, samples collected at nearby sites or close in time were more similar to each other than to other communities (Fig. S4 [16S] and Fig. S5 [18S]). Pairwise permutational multivariate analysis of variance (PERMANOVA) tests between sites showed significant differences between locations (e.g., LPL versus SDR1, *P* = 0.003) but not between sites at the same location (e.g., SDR1 versus SDR2, *P* = 0.604), suggesting a spatial influence on community composition (see the data on figshare at https://dx.doi.org/10.6084/m9.figshare.13653590). Pairwise beta-diversity

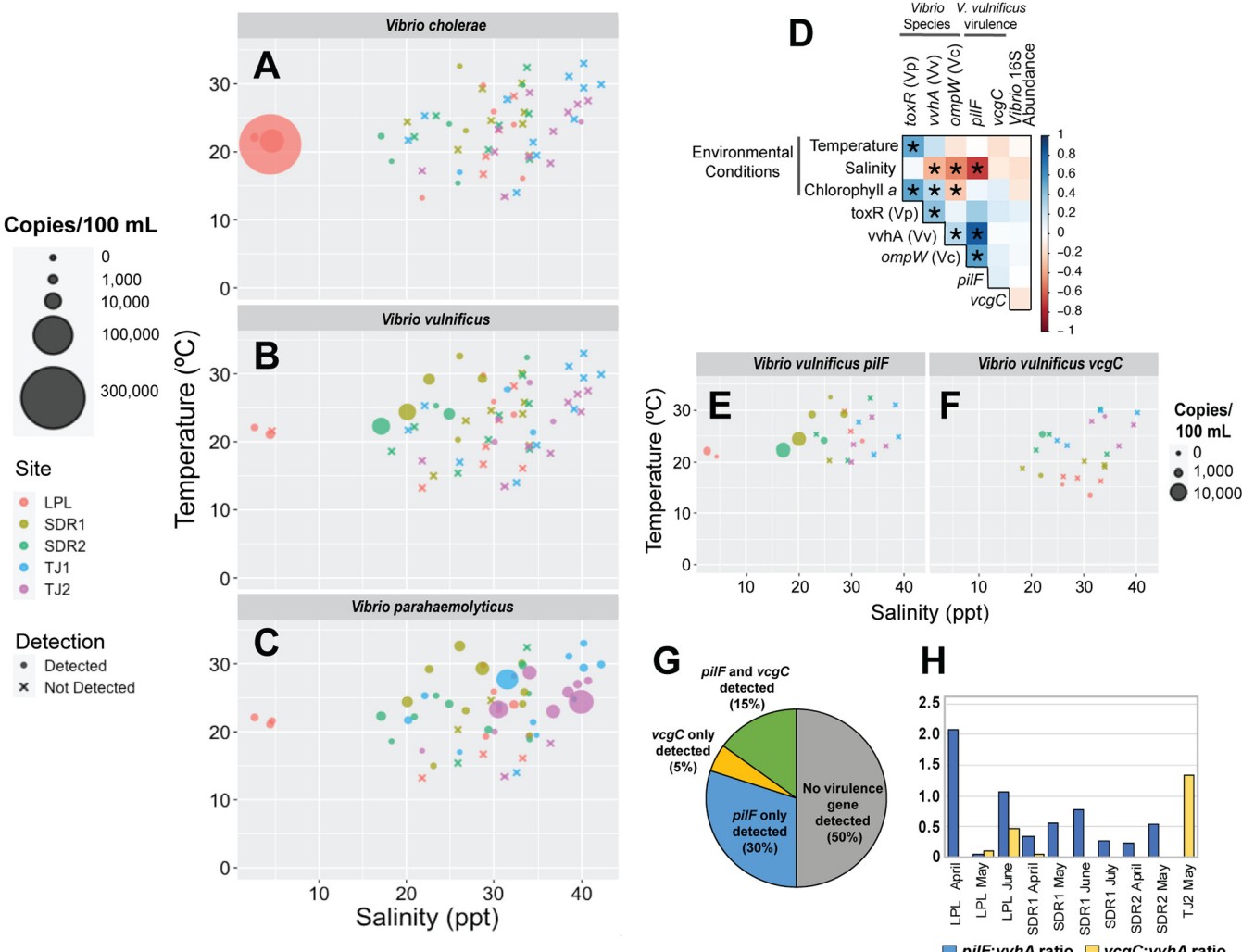

**FIG 3** Number of single-genome copy genes (a proxy for cell numbers) per 100 ml detected by digital droplet PCR. (A) The *Vibrio parahaemolyticus* (*Vp*) species-specific gene target *toxR*, (B) the *Vibrio vulnificus* (*Vv*) species-specific target *vvhA*, and (C) the *Vibrio cholerae* species-specific target *ompW*, with marker size corresponding to copy number and color corresponding to site, plotted against temperature and salinity. Data points marked with an "X" represent samples where that marker gene was not detected, also noted in the legend. (D) Spearman's rank correlation coefficients of associations between environmental conditions, the variables temperature, salinity, and chlorophyll *a*, and *Vibrio* species and virulence gene targets. Blue represents a strong positive correlation, while red represents a strong negative correlation, significant correlations ($P < 0.05$) are denoted with an asterisk (*). Number of copies detected per 100 ml by digital droplet PCR for the *Vibrio vulnificus* virulence-associated genes *vcgC* (E) and *pilF* (F) plotted against temperature and salinity. (G) The percentage of *V. vulnificus* samples where no virulence gene was detected, either *vcgC* or *pilF* was detected, or both were detected. (H) The ratio of the number of *pilF* and *vcgC* copies detected to total *V. vulnificus* determined by *vvhA* copy number.

plots also visually demonstrate this trend (Fig. S6 [16S] and Fig. S7 [18S]). Random forest classifiers predicted the overall accuracy to be ~92% for month-based classification and ~83% for site-based classification of samples in prokaryotic communities and 50% classification accuracy for both variables in eukaryotic communities (Fig. S2). The predictive accuracy for temperature ($R = 0.90$, mean square error [MSE] = 7.52) and salinity ($R = 0.85$, MSE = 23.85) was much better in the prokaryotic community than in eukaryotic communities (temperature, $R = 0.47$ and MSE = 25.55; salinity, $R = 0.44$ and MSE = 75.63), indicated by lower MSE and higher $R$ values in the former (Fig. S2).

**Abundance, distribution, and environmental drivers of pathogenic *Vibrio* species.** Pathogenic *Vibrio* species often cooccurred, and there was no site at which any of the three species went completely undetected (Fig. 3A to C). *V. cholerae* and *V. vulnificus* abundances were significantly associated with lower salinities (*V. cholerae*, $r = -0.48$ and $P < 0.001$; *V. vulnificus*, $r = -0.34$ and $P < 0.001$), while *V. parahaemolyticus* abundance was associated with higher temperatures ($r = 0.51$, $P = 0.014$) based on

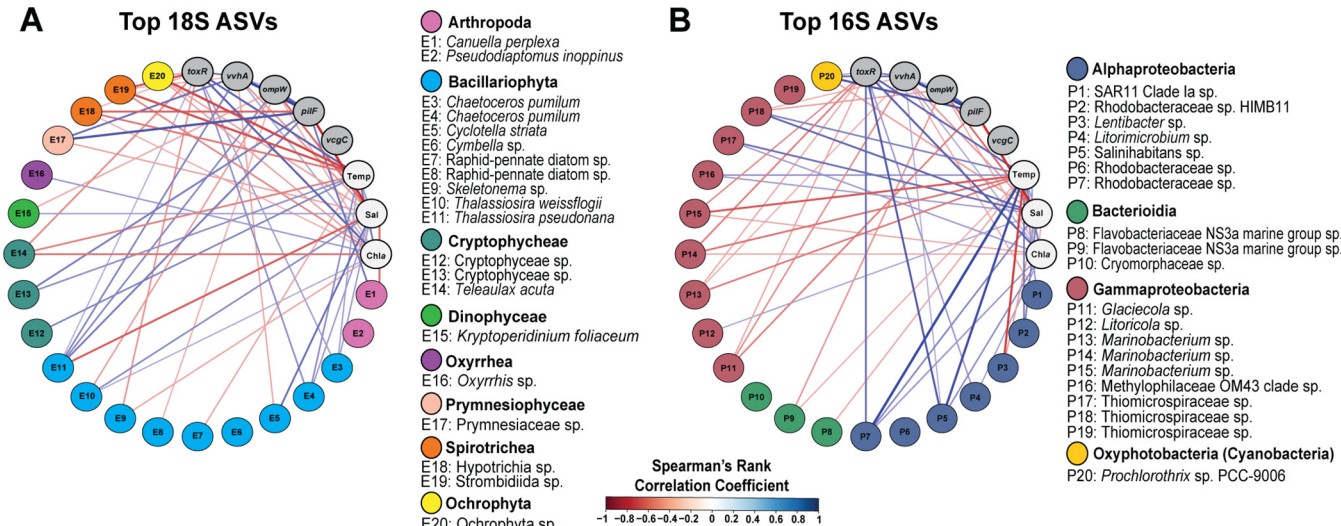

**FIG 4** Associations between pathogenic *Vibrio* species, environmental variables, and the 20 most abundant 16S and 18S ASVs. Spearman's rank correlations between the top 20 most abundant (A) eukaryotic and (B) bacterial and archaeal ASVs are organized by class, and only significant ($P < 0.05$) correlations are shown. Blue lines depict significant positive associations, while red lines represent negative associations, and line thickness indicates the strength of the correlation. Marker genes used to quantify pathogenic *Vibrio* species are shown in gray circles and correspond to the following species and virulence genes: *toxR*, *V. parahaemolyticus* species; *vvhA*, *V. vulnificus* species; *ompW*, *V. cholerae* species; *pilF*, *V. vulnificus* virulence-associated gene; *vcgC*, *V. vulnificus* virulence-associated gene.

Spearman's rank correlations (for primers and probes, see Table S1; for raw digital droplet PCR [ddPCR] copy numbers, see Table S2). High numbers of these species (>1,000 copies/100 ml) were only observed at temperatures of >20°C (Fig. 3A to D). The designation of "high abundance" for pathogenic *Vibrio* species was made based on previously published infectious dose estimates (24) and relative to samples in this study with lower vibrio concentrations. *V. parahaemolyticus* was detected in 80% (48/60) of samples and, while abundant at warm temperatures ($P < 0.05$) (Fig. 3D), it was more common at higher salinities than the other species (>40 ppt; Fig. 3C). Meanwhile, *V. vulnificus* and *V. cholerae* were both significantly associated with low salinity (Fig. 3A and B). Notably, high numbers of *V. cholerae* (~280,000 copies/100 ml) were detected at Los Peñasquitos Lagoon (LPL) from March to May, which occurred during a lagoon closure that occurs naturally each year and leads to an urban freshwater accumulation (Fig. 1F, Fig. 3A). *V. parahaemolyticus* reached high concentrations (~33,000 copies/100 ml) at the Tijuana River Estuary during a high-salinity (40 ppt) sampling point.

The *Vibrio vulnificus* genes *pilF* and *vcgC*, which are associated with human pathogenicity (25, 26), were also detected in the *V. vulnificus*-positive (20/60) samples. Both were most abundant at the SDR sites, with *pilF* reaching greater than 7,000 copies/100 ml (Fig. 3E and F). When *V. vulnificus* was detected, 50% of samples contained one or both of the virulence-associated genes (Fig. 3G). The ratio of these targets to *V. vulnificus* copies (potentially reflective of the number of virulence-associated gene copies per *V. vulnificus* cell) was often below 1, though sometimes closer to 2 as in the LPL April sample (Fig. 3H). Attempts to quantify the *V. parahaemolyticus* virulence-associated thermostable direct hemolysin gene (*tdh*) and TDH-related hemolysin gene (*trh*) using standard PCR and ddPCR were unsuccessful due to a lack of probe specificity (i.e., assay failure, not necessarily absence of targets in samples).

**Relationships between pathogenic *Vibrio* spp., planktonic community taxa, and the environment.** In addition to class-level taxonomic composition, we identified the top 20 most abundant ASVs in the eukaryotic, bacterial, and archaeal communities (Fig. 4) and analyzed their associations with both particular pathogenic *Vibrio* species and environmental variables based on Spearman's rank correlations. Relative abundance of the most abundant eukaryotic class, consisting of diatoms (class *Bacillariophyta*), was

mSystems®

positively associated with *V. parahaemolyticus* abundance and temperature, salinity, and chlorophyll *a* (Fig. S8), but individual diatom taxa exhibited genus- and species-level associations. For example, two *Chaetoceros pumilum* ASVs, representing the most abundant diatom taxa (Fig. 4A; see additional file 11 on figshare at https://dx.doi.org/10.6084/m9.figshare.13653359), and a *Thalassiosira pseudonana* ASV followed this class-level pattern of positive *V. parahaemolyticus* association, but an abundant *Skeletonema* ASV was negatively associated. The most abundant *T. pseudonana* ASV was positively associated with all three quantified *Vibrio* species, while a *T. weissflogii* ASV was associated with high salinity and low *V. cholerae*. Additionally, several copepod genera were positively linked to the *Vibrio* species found in lower salinity waters (*V. vulnificus* and *V. cholerae*) and the *V. vulnificus* virulence-associated gene *pilF* (see additional file 11 on figshare at https://dx.doi.org/10.6084/m9.figshare.13653359). This association does not appear to be driven by the most abundant copepod genus *Pseudodiaptomus*, which shows no correlations with target species or environmental variables at the ASV level (Fig. 4A). Rather, it appears to be related to several less abundant copepod genera found in low-salinity samples, including the genera *Canuella*, *Tigriopus*, *Sinocalanus*, and *Cyclops*, which are also linked to the pathogenic *Vibrio* spp. commonly found in lower salinity samples.

Within the bacterial and archaeal community, *V. parahaemolyticus* and *V. vulnificus* were positively associated with *Verrucomicrobia*, a bacterial phylum isolated from many sample types and nearly ubiquitous in the marine environment (27) (Fig. S8). *V. vulnificus*, *V. cholerae*, and the *V. vulnificus* virulence-associated gene *pilF* were positively associated with *Cyanobacteria* and negatively associated with *Campylobacter* spp., a pattern mirroring the negative association between these three marker genes and salinity. *V. cholerae* was negatively associated with *Bacteroidia* and *Kiritimatiellae*. However, associations based on genus and/or species were often not evident at the class level. For example, the *Gammaproteobacteria* and *Alphaproteobacteria* classes had no significant associations with any species or virulence genes. However, multiple gammaproteobacterial genera (e.g., *Glaciecola* and *Marinobacterium* genera) were associated with lower temperatures and lower concentrations of *V. parahaemolyticus*. In contrast, ASVs from alphaproteobacterial taxa, including *Rhodobacteraceae* and *Salinihabitans*, were associated with higher temperatures and higher *V. parahaemolyticus* concentrations (Fig. 4).

**Vibrio and associated viral diversity.** In addition to metabarcoding, a subset of the 60 total water samples was also used for culture-based enrichments of pathogenic vibrio bacteria (*N* = 23). These enrichments were conducted during the months of February, March, May, July, and August (CHROMagar Vibrio; see additional file 12 on figshare at https://dx.doi.org/10.6084/m9.figshare.13653359) and were derived from the same water samples collected to assess pathogenic *Vibrio* spp. abundance via ddPCR and community composition via 16S rRNA and 18S rRNA amplicon sequencing. A list of the samples collected and analyses conducted for each type of data can be found in additional file 13 on figshare at https://dx.doi.org/10.6084/m9.figshare.13653359. We conducted heat shock protein 60 (HSP60, also known as chaperonin protein 60 or cpn60) amplicon sequencing using DNA obtained from these enriched cultures to identify additional *Vibrio* species, including potential human pathogens and vibrio-associated viruses that were not identified by ddPCR or 16S amplicon sequencing. This revealed 557 unique *Vibrio* ASVs (see the data on figshare at https://dx.doi.org/10.6084/m9.figshare.13653512), of which 46 were abundant (>5% abundance across samples), indicating a high level of intraspecific diversity (see additional file 14 on figshare at https://dx.doi.org/10.6084/m9.figshare.13653359). These included *V. parahaemolyticus* and *V. cholerae* but not *V. vulnificus*, which was detected by ddPCR and shotgun sequencing but was absent in the HSP60 data set. Several *V. antiquarius* ASVs were abundant across the majority of samples, particularly in SDR and TJ sites, though these ASVs were entirely missing from the LPL May and August samples. Additional human and animal pathogens, including *V. furnissii*, *V. angularium*, and *V. metschnikovii*,

were detected in relatively low abundance. This diversity was confirmed and expanded upon with shotgun sequencing (see additional file 15 on figshare at https://dx.doi.org/10.6084/m9.figshare.13653359), which revealed even more *Vibrio* species, including all three target pathogens, the opportunistic human pathogen *V. alginolyticus*, and the animal pathogens *Vibrio anguillarum*, *V. ordalii*, *V. harveyi*, and *V. campbellii*, and *V. splendidus* (see the data on figshare at https://dx.doi.org/10.6084/m9.figshare.13653497). Putative vibrio-associated phages were also observed, including vB VpaM MAR, which was highly abundant in the TJ2-Feb sample (see the data on figshare at https://dx.doi.org/10.6084/m9.figshare.13653497), and the vibrio temperate phage VP882 was also observed.

**Vibrio genes associated with human pathogenicity.** To enhance our understanding of the pathogenic *Vibrio* spp. present in our samples, DNA obtained from the vibrio-enriched samples (N = 23) was shotgun sequenced and screened for additional human health-associated genes, including virulence and antibiotic resistance genes. We confirmed that all three species-specific ddPCR target genes (*toxR*, *ompW*, and *vvhA*) were present in the shotgun metagenomics data set (Fig. 5A). The *V. cholerae ctxA* gene (28) responsible for producing cholera toxin and causing the disease was not detected, nor were the accessory cholera enterotoxin (*Ace*) (29), the zona occludens toxin (*zot*) (30), and *tcpA* (31), which is required for *V. cholerae* host colonization. *V. cholerae rtxA* genes that are potentially virulence associated (32) and *vgrG* genes required for type VI secretion system (T6SS)-dependent cytotoxic effects of *V. cholerae* on eukaryotic cells (33) were detected. The *V. parahaemolyticus* virulence-associated genes *trh* and *tdh*, which have high sequence homology and are combined in this analysis, were detected in many samples in the metagenomic data despite unsuccessful ddPCR detection (Fig. 5A). The *V. vulnificus* virulence-associated gene *GbpA*, responsible for production of the *N*-acetylglucosamine (GlcNAc)-binding protein A and highly expressed in clinical *V. vulnificus* isolates, was also detected.

The most abundant antibiotic resistance genes detected across all samples were associated with resistance to glycopeptide and aminoglycoside antibiotics, tetracycline, and proteins involved in resistance to both cephalosporin and penam antibiotics (Fig. 5B). Furthermore, genes linked to transport and regulation of multiple drugs were also identified. Among the samples with high concentrations of pathogenic vibrio bacteria, as quantified by ddPCR (Table S2, appearing in bold on the axis of Fig. 5B), the LPL and SDR sites in May had relatively high abundances of several antibiotic classes compared to other samples, including carbapenem resistance at LPL and SDR2 and tetracycline and fluoroquinolone at SDR2.

## DISCUSSION

**Drivers and ecological relevance of *Vibrio*-plankton dynamics.** Microbial prokaryotes and eukaryotes play important functional roles in vibrio ecology. They provide attachment substrate and nutrition, facilitating environmental persistence, and can also potentially impact virulence. While functional characteristics of microbes may potentially be shared at high taxonomic levels (e.g., class), often, diversity at the level of genus or species dictates interactions with the environment and other organisms. We observed that associations between pathogenic *Vibrio* spp. and both prokaryotic and eukaryotic organisms were dependent on the level of taxonomic classification. Likewise, broad taxonomic groupings, such as class, did not capture potentially important ASV-level associations. For example, while the broad bacterial class *Gammaproteobacteria* showed no significant associations with pathogenic *Vibrio* species, ASVs identified at a genus level did (Fig. 4B). High-throughput sequencing is a useful tool for identifying these potential interactions and generating ecologically relevant hypotheses for future investigation.

A primary aim of our study was to assess pathogenic *Vibrio* spp. interactions with eukaryotes, which play an important role in both vibrio ecology and human health. Diatoms were the most abundant eukaryotic organisms in our samples (>28% of 18S sequences), and prior studies suggest that they are frequently associated with high *Vibrio* spp. concentrations (34–36). Individual diatom species also host distinct

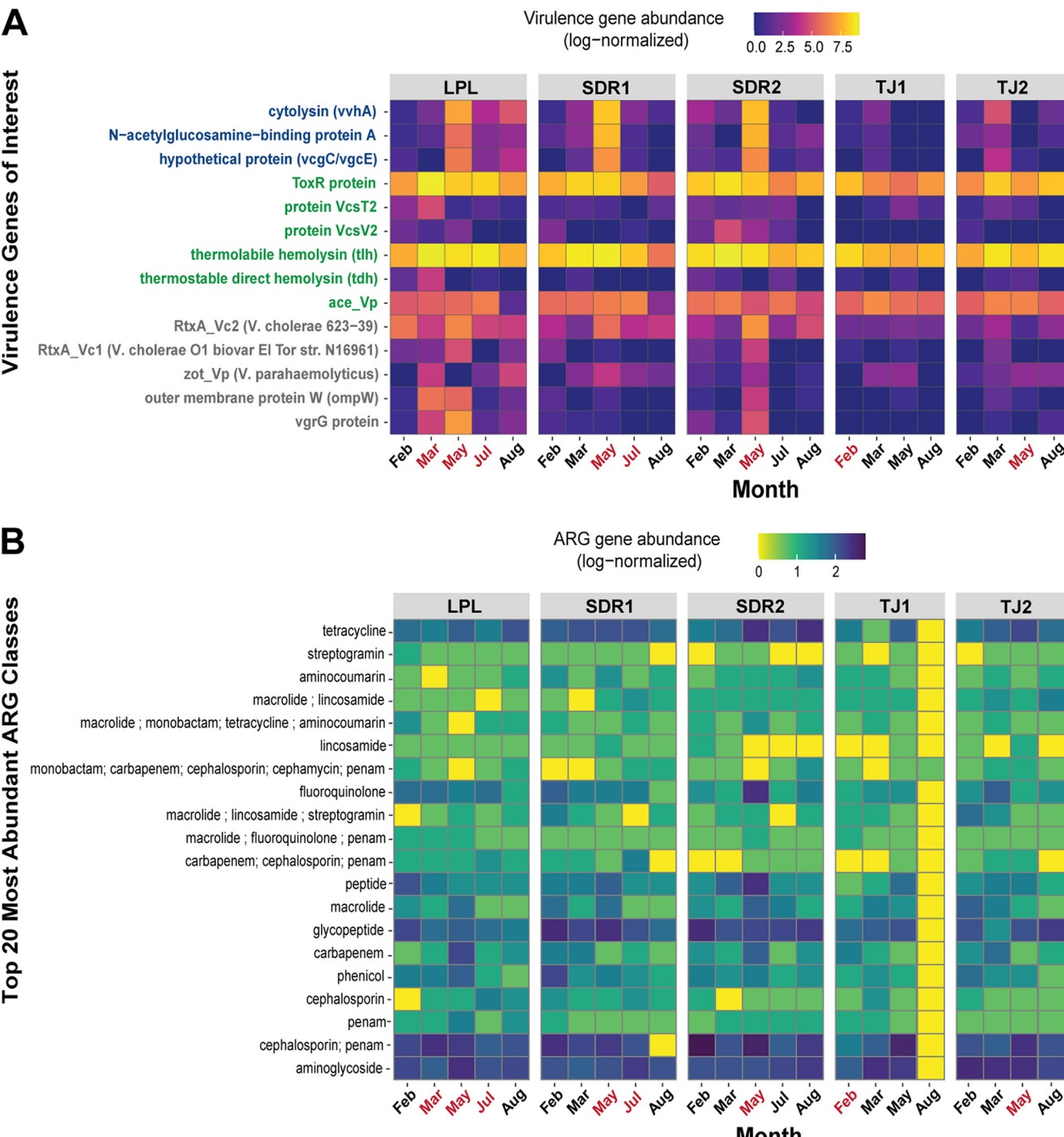

**FIG 5** Virulence and antibiotic resistance gene profiles of *Vibrio*-enriched communities based on shotgun metagenomic sequencing. (A) Heat map depicting log normalized abundance of genes associated with virulence identified in the literature. On the *y* axis, genes associated with *V. vulnificus* are in dark blue, genes associated with *V. parahaemolyticus* are in green, and *V. cholerae* are in gray. (B) Heat map showing log normalized abundance of the top 20 most abundant antibiotic resistance gene (ARG) classes as identified in the CARD database across months and sites. For panels A and B, months where high concentrations of one or more pathogenic *Vibrio* sp. were detected by ddPCR are indicated in red on the *x* axis.

microbial communities (21), release unique dissolved organic matter substrates (37–39), and have variable susceptibility to viral or bacterial infection (40–42). Additionally, two diatom genera (*Thalassiosira* and *Cyclotella*) can produce chitin, a polymer that stimulates attachment, horizontal gene transfer, and bacterial competition in *Vibrio* species (12, 15, 16, 43), and many other diatoms contain chitin synthesis

genes (19). The most abundant diatom ASVs observed in our samples were identified as *Chaetoceros pumilus*, which were linked to high levels of *V. parahaemolyticus*, high temperature, high salinity, and high chlorophyll *a* (Fig. 4). *Thalassiosira* diatoms were also abundant in our samples and were linked to different *Vibrio* spp. and environmental conditions. *T. pseudonana* was positively linked to all target species, while *T. weissflogii* was negatively correlated with *V. cholerae*. Additionally, a *Cyclotella striata* ASV was negatively associated with *V. vulnificus*, possibly due to their different environmental preferences (*C. striata* was associated with high salinity and *V. vulnificus* with low salinity). A prior laboratory study by Frischkorn et al. observed *V. parahaemolyticus* attaching to the chitin-producing diatom *T. weissflogii*, suggesting an unexplored mechanism of environmental persistence (44). While we observed no significant relationship between *T. weissflogii* and *V. parahaemolyticus* in our samples, attachment might occur with the closely related species *T. pseudonana*, which was positively associated with *V. parahaemolyticus*. Thus, *T. pseudonana* may actually be a more ecologically relevant model for studying these interactions. Future studies should investigate on a mechanistic level whether chitin production influences species-specific associations between pathogenic vibrios and diatoms.

Interactions between pathogenic *Vibrio* spp. and planktonic copepods are important, well-studied coastal phenomena with demonstrated human health implications, but, to our knowledge, have not been investigated using high-throughput sequencing. Laboratory and field studies have shown conflicting results that, as with diatoms, may be partially explained by functional differences resulting from insufficient taxonomic resolution. Environmental studies accounting for copepod taxonomy in a community context are rare and often qualitative (45, 46). We observed positive correlations between pathogenic *Vibrio* spp., particularly those found in lower salinities, and several copepod genera (see additional file 11 on figshare at https://dx.doi.org/10.6084/m9 .figshare.13653359). *Pseudodiaptamus inopinus*, an invasive species originating in Asia (47, 48), was not significantly associated with any *Vibrio* species but was highly relatively abundant during peak abundances of *V. cholerae* and *V. vulnificus* at LPL and the SDR sites and peak *V. parahaemolyticus* abundances at the TJ sites. The *Harpacticoid* genera *Canuella* and *Tigriopus* were positively associated with *V. vulnificus* and the virulence-associated gene *pilF*, which is notable since the type IV pilus (containing the *pilF* subunit) is involved in chitin attachment to *Vibrio* spp (14). *Tigriopus* was also positively associated with *V. cholerae*. *Tigriopus* is a well-established laboratory model genus with gene-silencing capabilities and full or partially assembled genomes for several species (49–51). Thus, *Tigriopus* and *Canuella* spp. may be candidate genera for future laboratory studies involving ecologically relevant *Vibrio*-plankton interactions.

*Vibrio* associations with individual planktonic taxa must be viewed in the context of shared ecological preferences. Positive associations between pathogenic *Vibrio* species and planktonic ASVs across a large number of diverse samples may suggest either common environmental drivers or actual interactions (e.g., mutualism, commensalism). These two possibilities are not mutually exclusive and are challenging to extrapolate, but this does not detract from relevant positive associations. For example, *V. vulnificus* and *V. cholerae* are associated with low salinities. They are positively associated with the diatom *Thalassiosira pseudonana*, which is also associated with lower salinities, suggesting a shared environmental preference. Whether these species are associated because they are actually interacting or simply cooccur under the same conditions, abundance in similar conditions increases potential interactions (whereas organisms not found together are unlikely to interact). Negative associations may represent antagonistic interactions or differing environmental niches; however, a lack of statistically significant association between organisms that are both abundant does not preclude interactions. For example, environmental factors may drive abundance of a *Vibrio* species, while grazing pressure may predominantly drive diatom abundance or community dominance. This would mask a correlation but not preclude important interactions of these organisms in the environment.

Associations despite different environmental preferences may suggest a direct ecological interaction, though further mechanistic studies would be needed to confirm these relationships. For example, *T. pseudonana* is positively associated with all three pathogenic *Vibrio* spp. but only shares the low-salinity association with *V. vulnificus* and *V. cholerae*; thus, additional interactions not related to shared environmental preferences may be occurring, including potential attachment and biofilm formation as discussed above. Furthermore, the time points at which samples are collected may influence associations observed. One limitation of our study is that in sampling monthly, we cannot understand the short-term dynamics (i.e., on scales of days or weeks) of vibrios and the planktonic community, including lagged relationships between taxa. Furthermore, the impacts of long-term climate patterns cannot be established. While our study establishes a baseline for the abundance and microbial ecology of pathogenic vibrios in this environment, future studies featuring high-resolution time-series or long-term monitoring would be extremely valuable.

**Species-specific environmental preferences of pathogenic *Vibrio* spp. suggest local risks and potential for future geographic expansion.** Here, we present the first quantification and ecological analysis of pathogenic *Vibrio* spp. in the Southern California coastal region, a potentially high-risk area due to warm coastal seawater temperatures, high residential and tourist water use, and recreational and commercial seafood harvesting. Given the southern location of San Diego in the United States, patterns of *Vibrio* spp. abundance observed here may be a preview for future vibrio distributions in adjacent colder waters as climate change increases global sea surface temperatures. *Vibrio* spp. infections in Southern California have increased in recent years (52), particularly in San Diego County; the most recent year assessed, 2018, showed the highest number of infections ever reported and an infection rate substantially higher than both the California and U.S. infection rates (53). The most common causes of these infections are the species *V. parahaemolyticus*, *V. alginolyticus*, *V. vulnificus*, *V. cholerae*, and other unidentified species (2, 53). We observed distinct environmental preferences among *V. cholerae*, *V. vulnificus*, and *V. parahaemolyticus* related to salinity and temperature (Fig. 3A to C). While these environmental factors are known to drive *Vibrio* distribution (17), many studies focus on single species or the *Vibrio* genus as a whole, overlooking species shifts in response to surrounding environmental community changes. By quantifying all three species, we capture some of these dynamics.

All three species reached highest abundances above 20°C, a temperature at which human *Vibrio* infections become a serious concern (3, 54, 55) (Fig. 3). *V. parahaemolyticus*, which is the most common cause of both San Diego and U.S. infections, particularly due to contaminated seafood, was significantly associated with higher temperatures ($r = 0.51$, $P = 0.014$). Peak abundances were detected in both moderate and very high (>40 ppt) salinities that were outside the range reported in previous studies (Fig. 3A), which may suggest unique high-salinity adaptations in these populations. A review by Takemura et al. noted that in contrast to other pathogenic species, *V. parahaemolyticus* typically occupies a broader salinity range of 3 to 35 ppt and a warmer, more narrow temperature range (17, 54, 55). Thus, we hypothesize that in these populations, halotolerance may enable *V. parahaemolyticus* to take advantage of ideal temperature conditions.

While *V. cholerae* and *V. vulnificus* were also most common in warm conditions, their abundance was significantly associated with lower salinities (*V. cholerae*, $r = -0.48$ and $P < 0.001$; *V. vulnificus*, $r = -0.34$, $P < 0.001$), which suggests that rising seawater temperatures combined with urban and watershed-associated freshwater runoff may increase risk of these species in Southern California and close geographic regions. This association with low salinity is in agreement with previous studies; while *V. cholerae* has been reported in high-salinity conditions, it is most common in low salinities, hence its tendency to contaminate drinking water, and *V. vulnificus* grows poorly at salinities higher than 25 ppt, preferring the range of 10 to 18 ppt (54, 56). Both species peaked during warm summer months, typically 1 to 2 months before peak temperature, and high abundances (relative to the samples in this study and infectious dose

estimates, e.g., [56]) were only found from March to July (see the data on figshare at https://dx.doi.org/10.6084/m9.figshare.13653512).

**Diverse *Vibrio* populations contain genes associated with virulence and antibiotic resistance.** As targeted sequencing analyses, ddPCR and amplicon-sequencing methods are limited in their ability to assess the genetic potential and possible pathogenicity of vibrio populations; thus, we supplemented these data with shotgun sequencing of vibrio-enriched cultures derived from the same samples. While this method also has limitations (e.g., some species may grow faster or slower than others, or some may not grow at all on selective medium), it enables a nontargeted and broad analysis of the vibrio species present, which we use to improve our understanding of their diversity and potential risk to human health in the context of the ecological associations we discuss above.

We observed *Vibrio* species, beyond the target human pathogens, that may be a concern for human or animal health. We observed *V. alginolyticus*, which is responsible for many human infections. Additionally, a common member of the vibrio community *V. antiquarius* (formerly called *Vibrio* sp. Ex25, which is the MetaPhlAn2 annotation) coincided with high abundances of *V. parahaemolyticus* (see additional file 14 on figshare at https://dx.doi.org/10.6084/m9.figshare.13653359) in diatom-dominant eukaryotic communities. This species, originally isolated from deep-sea hydrothermal vents (57), is closely related to *V. parahaemolyticus* and *V. alginolyticus* and may possess factors involved in human disease caused by coastal *Vibrio* spp. Since both recreational and commercial seafood harvesting are popular in Southern California, animal-associated vibrio pathogens detected in our samples, including *Vibrio anguillarum*, *V. ordalii*, *V. harveyi*, and *V. campbellii*, and *V. splendidus* (see the data on figshare at https://dx.doi.org/10.6084/m9.figshare.13653497) may also require further investigation.

*Vibrio* phages identified by shotgun sequencing in this study may be potential candidates for phage therapy, which can reduce pathogenic *Vibrio* species in aquaculture (58, 59). For example, ~82% of sequences from the TJ2-Feb sample belonged to vibrio phage vB VpaM MAR (see additional file 15 on figshare at https://dx.doi.org/10.6084/m9.figshare.13653359 and data at https://dx.doi.org/10.6084/m9.figshare.13653497), while the remaining sequences belonged primarily to *V. parahaemolyticus* and *V. EX25* (also known as *V. antiquarius* [58]), which may suggest that one of these species is the phage host. The vibrio temperate phage VP882 was also observed, which was originally isolated from a pandemic *V. parahaeomolyticus* O3:K6 strain shown to lyse *V. parahaemolyticus*, *V. vulnificus*, and *V. cholerae* strains (60).

The presence of virulence-associated genes increases the potential for *Vibrio* infection, particularly as they can be horizontally transferred among species in the community (60, 61). A high percentage of *V. vulnificus*-positive samples contained virulence-associated genes as measured by ddPCR (Fig. 3G and H), with half containing one or both of the virulence-associated genes *vcgC* or *pilF*. Along the North Carolina coast, Williams et al. found that 5.3% of the *V. vulnificus* examined possessed the *vcgC* gene, while *pilF* was detected in 45% of samples. Shotgun sequencing revealed additional virulence-associated genes present in the *Vibrio* populations, including the *V. parahaemolyticus* genes *trh* and *tdh*, which are challenging to detect and quantify due to high sequence variability (61). Future studies could utilize these sequences to region-specific primers.

Pathogenic species of *Vibrio* bacteria can also harbor multiple antibiotic resistance genes (62, 63), which, like virulence genes, can be transmitted between strains and even species via horizontal gene transfer. In previous studies, isolates of *V. vulnificus* have been shown to be resistant to eight or more antibiotics (64), with similar resistance profiles in virulent and nonvirulent strains. Several antibiotic resistance gene classes were present in our study, with many evenly distributed across sites, suggesting potential widespread antibiotic resistance in local *Vibrio* populations. Two sites with high levels of all three pathogenic species, LPL May and SDR2 May, have relatively high levels of many different antibiotic classes, suggesting that these strains may be highly antibiotic resistant. Pairing these data with the abundance of virulence genes is

a useful tool for understanding what populations may be dangerous and, when paired with the planktonic community data, understanding which other organisms may be serving as vectors or reservoirs for these strains in the environment.

**Conclusions.** Since many *Vibrio* species in the marine environment are known human pathogens, it is critical to understand their environmental preferences and temporal patterns, ecological interactions, and genetic potential in the interest of public health. We observed that pathogenic *Vibrio* species exhibited unique temperature and salinity preferences and were part of a diverse *Vibrio* community that harbored both antibiotic resistance genes and genes associated with virulence. Pathogenic species were associated with specific eukaryotes, such as diatoms and copepods, and these relationships may facilitate attachment and environmental persistence. Relationships between pathogenic *Vibrio* spp. and both prokaryotic and eukaryotic community members in general were often ASV-specific, suggesting that associations based on higher-level taxonomic classifications may mask important interactions and impede ecologically relevant laboratory studies. Furthermore, we identified shared environmental conditions that correlate with high levels of pathogenic vibrios and potential environmental reservoirs, which should be taken into consideration in developing a predictive understanding of pathogenic vibrio ecology.

## MATERIALS AND METHODS

**Environmental sampling and *Vibrio* enrichment.** Monthly water sampling was conducted from December 2015 to November 2016 at three locations in San Diego County: Los Peñasquitos Lagoon (LPL), the San Diego River (SDR), and the Tijuana River Estuary (TJ) (Fig. 1A to D). For intrasite comparisons, two different sites at SDR (SDR1 and SDR2) and TJ (TJ1 and TJ2) were sampled, totaling five sampling sites. This resulted in a total of 60 sampling points. Temperature and salinity were measured at each sampling point between 12 p.m. and 1 p.m. using a YSI Pro 30 field instrument (YSI Inc.), and unfiltered water samples were collected in 4-L opaque bottles and processed in the lab beginning no more than 2 h after collection. These samples were kept in a cool area at roughly room temperature rather than at 4°C to prevent a viable but nonculturable (VBNC) state in *Vibrio* bacteria (65). Water samples collected at all 60 time points were utilized for *Vibrio* spp. quantification and amplicon sequencing, while a subset of these same water samples ($N = 23/60$) was also used for additional vibrio enrichment and sequencing protocols (see below).

To determine chlorophyll *a* concentrations and extract nucleic acids, water samples from all sampling points ($N = 60$) were gently filtered and flash frozen in the lab for downstream processing, and volume was determined based on sample turbidity; we collected as much volume on filters as possible before the filters clogged. For chlorophyll *a* quantification, 10- to 100-ml samples were collected on glass fiber filters with a 0.7-$\mu$m pore size (Whatman) and stored at −20°C. Samples were later extracted in 90% acetone overnight and measured on a 10-AU fluorometer (Turner), followed by addition of HCl and remeasurement to account for the chlorophyll *a* degradation product pheophytin (66). For downstream nucleic acid extractions, 50- to 400-ml samples were filtered onto 0.4-$\mu$m polycarbonate filters (Whatman), flash frozen in liquid nitrogen, and stored at –80°C until processing.

For a subset of the water samples noted above ($N = 23$ out of 60 sampling points), additional water was filtered, and these filters were plated on vibrio-selective medium to enrich for *Vibrio* spp. The samples used for this vibrio enrichment and sequencing protocol were collected during the months of February, March, May, July, and August. We filtered the maximum amount of seawater possible before filters clogged (1 to 100 ml) onto 0.45-$\mu$m sterile cellulose nitrate filter membranes, which were then lifted with tweezers from the filtration funnels, transferred to CHROMagar Vibrio (CHROMagar Microbiology) plates, and incubated overnight at 37°C. These communities formed colonies, which were then resuspended by adding 1 ml of either LB broth (Amresco) or Zobell marine broth 2216 (HiMedia), depending on sample salinity, using a sterile spreader to detach colonies from the plate and then gently pipetting the colonies and liquid using a 1-ml pipette. The liquid colony resuspensions were then frozen as 15% glycerol stocks at −80°C. Half of each glycerol stock was pelleted and used for downstream DNA extraction.

Water samples collected at all sampling points ($N = 60$) were used to quantify pathogenic *Vibrio* spp. using ddPCR and to conduct 16S and 18S amplicon sequencing, and these data were subsequently used for the correlation analyses with environmental variables (see below). Additionally, the subset of these same water samples ($N = 23$; see additional file 13 on figshare at https://dx.doi.org/10.6084/m9.figshare .13653359) that was used to enrich for *Vibrio* bacteria was subsequently used for HSP60 amplicon sequencing and shotgun metagenomic sequencing to characterize additional vibrio species and determine the presence of genes associated with human health (i.e., virulence and antibiotic resistance genes). Data derived from these vibrio-enriched cultures were not used in any of the presented correlation analyses.

**DNA and RNA extraction and cDNA synthesis.** Nucleic acids were extracted from filter samples using the NucleoMag plant kit (Macherey-Nagel) for genomic DNA (gDNA) and the NucleoMag RNA kit (Macherey-Nagel) for RNA. Initial sample lysis buffer resuspension and vortexing was completed

manually, the remainder using an epMotion liquid handling system (Eppendorf). gDNA was quantified using the Quant-iT PicoGreen double-stranded DNA assay kit (Invitrogen), and RNA was quantified using the Quant-iT RiboGreen RNA assay kit. Nucleic acid integrity was confirmed using an Agilent 2200 TapeStation (Agilent). RNA was reverse-transcribed into cDNA using the SuperScript III first-strand cDNA synthesis system (Invitrogen). Genomic DNA was extracted from *Vibrio*-enriched pellets using a DNeasy blood and tissue kit (Qiagen), with subsequent quantification and quality control as described above.

***Vibrio* digital droplet and endpoint PCR.** Digital droplet PCR (ddPCR) was performed to quantify single copy number genes specific to each species in known filtration volumes using previously designed quantitative PCR (qPCR) primers (Table S1 in the supplemental material) (25, 67–70, 92). Select pathogenic *Vibrio* species and virulence genes were quantified using the QX200 ddPCR system (Bio-Rad), following the manufacturer's protocols and recommended reagents. DNA was used as a template for the ddPCR as *Vibrio* species abundance was determined using genes with only a single copy per genome, and measuring the expression of genes with various copy numbers (i.e., using an RNA template) would invalidate the use of these data for quantification analysis. ddPCR assays were performed following manufacturer's instructions and previously published procedures detailed in Cao et al. (71), including running temperature gradients for each target to establish optimum reaction temperature. Primers and probes were ordered from Integrated DNA Technologies. Results from technical replicates were merged for analysis, and more than 19,000 droplets were measured per sample. Target-specific gBlocks (Integrated DNA Technologies) were used as positive controls for all ddPCR and endpoint PCR targets.

Single copy number gene targets for the species *V. parahaemolyticus*, *V. vulnificus*, and *V. cholerae* were quantified and used to approximate cell number per 100 ml of sample. We targeted *toxR* for *V. parahaemolyticus* (23), *vvhA* for *V. vulnificus* (24), and *ompW* for *V. cholerae* (25). *V. cholerae ompW* was not quantified during the months of December and April at SDR1 due to technical problems.

We also quantified the virulence-associated *V. vulnificus* genes *vcgC* (21) and *pilF* (22) in all samples where this species was detected. *PilF* is a protein required for pilus type IV assembly, and the particular polymorphism detected in our study and in prior qPCR-based studies is strongly associated with *V. vulnificus* human pathogenicity (26). The *vcgC* sequence quantified in our study was derived from a randomly amplified polymorphic DNA (RAPD) PCR amplicon associated with clinical isolates of *V. vulnificus*, which differs from the *vcgE* variant primarily found in environmental samples and is commonly used to inform whether a *V. vulnificus* strain has pathogenicity potential (71). Thus, detection of this gene in our study samples may be indicative that certain strains in the population may have the capacity to infect humans.

We attempted to quantify the *V. parahaemolyticus* virulence-associated thermostable direct hemolysin gene (*tdh*) and TDH-related hemolysin gene (*trh*) using standard PCR and ddPCR. These attempts were unsuccessful; however, we were able to collect data on the abundance and distribution of these genes using shotgun sequencing.

**High-throughput library preparation and sequencing.** Amplicon libraries were constructed and sequenced using either a cDNA or DNA template derived from filtered samples. For the 16S and 18S amplicons, duplicate libraries (derived from two separate filters) were constructed using cDNA in order to characterize biologically active community members for a total of 120 libraries sequenced per amplicon. For HSP60 analysis, single libraries were similarly constructed using DNA extracted from the *Vibrio*-enriched communities ($N = 23$, a subset of the $N = 60$ total sampling points; see additional file 13 on figshare at https://dx.doi.org/10.6084/m9.figshare.13653359). Libraries were sequenced at either the Institute for Genomic Medicine (IGM, University of California, San Diego) or at the University of California Davis Genome Center (https://dnatech.genomecenter.ucdavis.edu/), with 300-bp paired-end sequencing for the 16S and HSP60 amplicons (MiSeq reagent kit v3) and 150-bp paired-end sequencing for the 18S amplicon (MiSeq reagent kit v2).

The 16S rRNA gene small subunit (SSU-rRNA) V4-5 region was targeted to characterize the bacterial and archaeal community using primers 515F and 926R (72). The V9 region of the 18S rRNA gene was targeted for eukaryotic community composition using primers 1389F and 1510R (73). The universal region of heat shock protein 60 (HSP60), also known as chaperonin 60 (cpn60), was amplified and sequenced as described in Jesser and Noble 2018 using primers identified in previous studies (22, 74, 75). Sequences were filtered for quality using bbduk (65). Due to the size of the amplicon (549 to 567 bp), paired-end reads did not merge sufficiently for a robust analysis, so we used only high-quality forward reads.

For shotgun metagenomics libraries, DNA extracted from *Vibrio*-enriched communities was fragmented to 400 bp on an E210 sonicator (Covaris). Sequencing libraries were prepared using the NEBNext Ultra II DNA library prep kit (New England Biolabs), combined into two equimolar concentration pools of 13 samples each, and sequenced on an Illumina HiSeq4000 at the University of California Davis Genome Center with 250-bp paired-end reads.

**Bioinformatic and statistical analyses.** Demultiplexed sequences were analyzed using the QIIME 2 (version 2019.4) (76, 77) pipeline, and additional analyses and visualizations were conducted using the R package phyloseq (version 1.26.1) (78). Sequences were quality filtered, chimeric sequences were removed, and exact amplicon sequence variants (ASVs) (79) were defined using dada2 (80), with a maximum expected error threshold of 2.0 (default) for 16S and 18S rRNA gene amplicons and 5.0 for the HSP60 amplicon. For 16S and 18S amplicons, replicate samples were merged using the "qiime feature-table group" function. Taxonomy was assigned using Silva (81) version 132 for bacterial and archaeal 16S sequences, using PR2 (version 4.11) (82) for 18S sequences, and using the cpn60 database (75) with taxonomic designations derived from NCBI for HSP60 sequences, as described in Jesser and Noble 2018 (22). Chloroplast, mitochondrial, and eukaryotic sequences were removed from 16S data sets prior to downstream analyses. Alpha and beta diversity metrics for community composition were calculated using phyloseq.

*Alpha diversity analyses.* Singleton ASVs were retained for alpha diversity analyses. To examine alpha diversity, we calculated a number of alpha diversity metrics, focusing on the number of observed ASVs (an indicator of species richness) and Shannon diversity (an indicator of both richness and evenness). We then statistically compared alpha diversity of the categorical groups site and month by performing Kruskal-Wallis tests and investigated the relationship between the continuous variables temperature, salinity, and chlorophyll *a* and alpha diversity by performing Spearman's rank correlations.

*Beta diversity analyses.* Beta diversity-based analyses were conducted after filtering out taxa that were not observed at least three times in 20% of samples, transforming to an even sampling depth, and calculating Bray-Curtis dissimilarity matrices. To statistically evaluate relationships between metadata variables and community dissimilarity, we conducted PERMANOVA tests targeting both categorical and continuous variables using the VEGAN function Adonis (version 2.5.6) (83) to assess the predictive power of environmental variables on community composition. For the categorical variables, we preceded PERMANOVA analyses with a beta dispersion permutation betadisp test and principal-coordinate analysis (PCoA) visualization to determine whether dispersion of samples within each group was homogenous and to calculate distance to the centroid of each group cluster (see Materials and Methods for additional information). We conducted pairwise PERMANOVA tests between groups to better characterize spatial and temporal patterns. Additionally, we plotted pairwise sample dissimilarity values by site and month to quantitatively observe patterns in sample similarity (Fig. S4, Fig. S5).

PERMANOVA tests between months showed that temporally closer months were not significantly different from each other (e.g., March versus April, $P = 0.354$; March versus May, $P = 0.203$), and winter months in particular clustered together in the beta dispersion plots (Fig. S6, Fig. S7). In examining the effect of site on bacterial and archaeal (16S) community diversity, the beta dispersion permutation test for homogeneity of multivariate dispersions was significant (see data on figshare at https://dx.doi.org/10 .6084/m9.figshare.13653590), indicating that the dispersions were not homogenous across groups. A Tukey's honestly significant difference (Tukey HSD) test revealed that this was due to a significant difference in dispersion between sites SDR2 and LPL, while all other sites had similar dispersion.

Supervised learning in the form of a RandomForestClassifier estimator method was implemented using the QIIME 2 (version 2019.4) plug-in "sample-classifier" to predict categorical metadata variables (i.e., site, month) in response to community composition, while a RandomForestRegressor estimator was employed to predict continuous metadata variables (i.e., temperature, salinity). Default values were used for both algorithms, along with a random seed generator (–p-random-state 123) to replicate results each run.

For taxonomic analysis, shotgun sequences were first quality filtered using trimmomatic (84) and checked for quality using FastQC (Babraham Bioinformatics, https://www.bioinformatics.babraham.ac.uk/ index.html). We then assigned taxonomy to unassembled reads using MetaPhlAn2 (85) (default parameters), which relies on clade-specific marker genes identified from ~17,000 reference genomes, and visualized the output using GraphPhlAn (86) (using export2graphlan.py converter: –most_abundant 100 – abundance_threshold 1 –least_biomarkers 10 –annotations 5,6 –external_annotations 7 –min_clade_size 1). In order to analyze and target the virulence genes and antibiotic resistance genes, reads were first assembled into sample-specific assemblies using clc-assembler (CLC-Assembly-Cell version 5.1.1.184548) with the following parameters: "–wordsize 31 –paired fb es 0 700 –min-length 200". These were then grouped and merged by site into a global assembly, utilizing CD-HIT (version 4.6) (87) to remove subsequences/duplicates with a minimum alignment identity of 0.95 and minimum contig length of 300 bp for merging assemblies. Open reading frames (ORF) were called on all contigs using FragGeneScan (version 1.31), and read counts for each ORF for all samples were obtained by mapping all reads to predicted ORFs using clc_mapper (CLC Read Mapper - Version 5.1.1.184548). Subsequently, all mapped read counts were merged across samples for all ORFs.

In order to screen for the virulence genes, representative sequences for these genes were queried against the global assembly using BLAST (version ncbi-blast+-2.4.0) (88). In order to screen for the virulence genes, sequences for these genes were queried against all ORFs generated from the global coassembly using BLASTN (version ncbi-blast+-2.4.0) (88). Only those ORFs that had greater than 80% percent identity and greater than 75% query coverage to the virulence genes were selected. Mapped read counts for these virulence gene-coding ORFs across all samples were then used to visualize gene abundance results. Additionally, resistance gene identifier (RGI) was used against the CARD database (89) to predict the presence of antibiotic resistance genes based on homology and single-nucleotide polymorphism (SNP) models in site-specific assemblies. All resulting .json files were parsed to collate all hits across all genomes in a matrix-like format. All count data from each of these analyses were log transformed, manipulated using "dplyr" (version 1.0.1) and "reshape2" (version 1.4.4), and visualized using "ggplot2" (version 3.3.2) to generate the plot in R (version 4.0.2).

Spearman's rank correlation coefficients were calculated to explore relationships between environmental variables, *Vibrio* quantification data, and relative abundance of groups of interest in the amplicon sequencing data. Correlations were visualized as correlograms using the corrplot package in R (90) or as network graphs using the R package qgraph (91).

**Data availability.** The sequencing data sets generated and/or analyzed during the current study are available in the NCBI repository (BioProject accession no. PRJNA593265; BioSample accession no. SAMN13474661-SAMN13474785, SAMN13475110-SAMN13475236, and SAMN13475238-SAMN13475318). Scripts for analyses are in additional file 16 on figshare at https://dx.doi.org/10.6084/m9.figshare.13653362 and additional file 17 on figshare at https://dx.doi.org/10.6084/m9.figshare.13653353. The metadata file used for QIIME 2 and phyloseq analyses, which also includes environmental metadata, is also available on figshare at https://dx.doi.org/10.6084/m9.figshare.13653593.

## SUPPLEMENTAL MATERIAL

Supplemental material is available online only.

**FIG S1**, PDF file, 0.1 MB.
**FIG S2**, PDF file, 0.2 MB.
**FIG S3**, PDF file, 0.1 MB.
**FIG S4**, PDF file, 0.1 MB.
**FIG S5**, PDF file, 0.1 MB.
**FIG S6**, PDF file, 0.1 MB.
**FIG S7**, PDF file, 0.1 MB.
**FIG S8**, PDF file, 0.5 MB.
**TABLE S1**, PDF file, 0.1 MB.
**TABLE S2**, PDF file, 0.1 MB.

## ACKNOWLEDGMENTS

We acknowledge Rachel Noble and Brett Froelich for project guidance and assistance in culturing, identifying, and screening *Vibrio* bacteria and Kelsey Jesser for assistance in analyzing HSP60 amplicon sequences. We further acknowledge Lucy Mao, Meredith Raith, and Amanda Thygerson for assistance with ddPCR and sample collection. We thank Holly Lutz and Sarah Allard for help in editing and improving the manuscript.

Funding was provided to R.E.D. by a San Diego Institutional Research and Academic Career Development Award (IRACDA) (NIH/NIGMS IRACDA K12 GM068524), the National Science Foundation postdoctoral research fellowship in biology (PRFB; P2011025), the University of California, San Diego, Center for Microbiome Innovation, and the Achievement Rewards for College Scientists (ARCS) Foundation. This study was supported, in part, by National Science Foundation (NSF-OCE-1637632 and NSF-OCE-1756884), National Oceanic and Atmospheric Administration (NOAA) (NA15OAR4320071 and NA19NOS4780181), and Gordon and Betty Moore Foundation (GBMF3828) grants to A.E.A.

R.E.D., A.E.A., J.A.S., and J.F.G. were responsible for the conception and design of the study. R.E.D., H.Z., and A.R. were responsible for acquiring and analyzing study data, and R.E.D. and A.E.A. were responsible for data interpretation.

The authors declare that they have no competing interests.

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
