## [Reviewer comments · mSystems]

Pathogenic *Vibrio* species are associated with distinct environmental niches and planktonic taxa in Southern California (USA) aquatic microbiomes

Rachel Diner, Drishti Kaul, Ariel Rabines, Hong Zheng, Joshua Steele, John Griffith, and Andrew Allen

Corresponding Author(s): Andrew Allen, J Craig Venter Institute / University of California San Diego

Review Timeline:

Submission Date:

May 17, 2021

Accepted:

June 8, 2021

Editor: Hans Bernstein

Reviewer(s): Disclosure of reviewer identity is with reference to reviewer comments included in decision letter(s). The following individuals involved in review of your submission have agreed to reveal their identity: Lauren Marie Seyler (Reviewer #2)

Transaction Report:

DOI: <https://doi.org/10.1128/mSystems.00571-21>

June 8, 2021

Dr. Andrew E Allen
J Craig Venter Institute / University of California San Diego
La Jolla

Re: mSystems00571-21 (Pathogenic *Vibrio* species are associated with distinct environmental niches and planktonic taxa in Southern California (USA) aquatic microbiomes)

Dear Dr. Andrew E Allen:

Your manuscript has been accepted, and I am forwarding it to the ASM Journals Department for publication. For your reference, ASM Journals' address is given below. Before it can be scheduled for publication, your manuscript will be checked by the mSystems senior production editor, Ellie Ghatineh, to make sure that all elements meet the technical requirements for publication. She will contact you if anything needs to be revised before copyediting and production can begin. Otherwise, you will be notified when your proofs are ready to be viewed.

We recognize that the video files can become quite large, and so to avoid quality loss ASM suggests sending the video file via <https://www.wetransfer.com/>. When you have a final version of the video and the still ready to share, please send it to Ellie Ghatineh at eghatineh@asmusa.org.

Thank you for submitting your paper to mSystems.

Sincerely,

Supplemental Material: Accept
Supplemental Material: Accept
Supplemental Material: Accept
Supplemental Material: Accept
Supplemental Material: Accept
Supplemental Material: Accept
Supplemental Material: Accept
Supplemental Material: Accept
Supplemental Material: Accept
Supplemental Material: Accept